# Recognition of *Davidsoniella virescens* on *Fagus sylvatica* Wood in Poland and Assessment of Its Pathogenicity

**DOI:** 10.3390/jof10070465

**Published:** 2024-06-29

**Authors:** Tadeusz Kowalski, Piotr Bilański

**Affiliations:** Department of Forest Ecosystems Protection, University of Agriculture in Krakow, Al. 29 Listopada 46, 31-425 Krakow, Poland; rltkowal@cyf-kr.edu.pl

**Keywords:** *Davidsoniella virescens*, wood stain, sapstreak, invasive fungi, quarantine pests, phytosanitary risk

## Abstract

*Davidsoniella virescens* is so far only known in North America. However, recently in southern Poland, blackish growth consisting of fungal mycelia and sporulation structures was found on the wood of *Fagus sylvatica*. As a result of isolation, 17 cultures of this fungus were obtained. All cultures produced an intense sweet odor. This fungus, both in situ and in vitro, abundantly produced perithecia with long necks and asexual stage. Particularly characteristic was the production of variable endoconidia in two types of phialophores differing mainly in the width of the collarette. The nucleotide sequences for five gene fragments of representative cultures were used in phylogenetic analyses: 18S; the internal transcribed spacer regions ITS1 and ITS2, including the 5.8S gene (ITS); 28S region of the ribosomal RNA (rRNA), β-tubulin 2 (TUB2) and translation elongation factor 1-α (TEF1). Based on morphological and phylogenetic analyses, the fungus on European beech in Poland was identified as *Davidsoniella virescens*. The optimal temperature for radial colony growth was 20 °C. However, the differences between colony diameter at 25 °C compared to that at the optimal temperature were not statistically significant. Six *D. virescens* isolates were used for pathogenicity assay. They were inoculated into wounds on stems of two-year-old seedlings of *Fagus sylvatica* and *Acer saccharum* (36 seedlings of each tree species). Final evaluation was performed 4 months after inoculation. No external symptoms were observed in any *A. saccharum* seedling, neither in the crown nor on the stem. However, 13.9% of *F. sylvatica* seedlings showed wilting symptoms throughout the entire crown within 3–6 weeks after inoculation. Moreover, after 4 months on the stems of 30.6% beech seedlings, necrotic lesions with a length of 1.3 to 7.2 cm were formed, without any symptoms of wilting. The most noticeable internal symptom was the discoloration of the wood, which was observed in all inoculated seedlings of both tree species. All *D. virescens* isolates caused greater wood discoloration in *F. sylvatica* than in *A. saccharum*. Most of the differences found in the extent of discoloration between host plants were statistically significant. The discoloration caused by all *D. virescens* isolates in *F. sylvatica* was significantly greater than in the control. However, none of the isolates tested on *A. saccharum* caused significantly greater wood discoloration compared to the control. Pathogenicity tests showed that the *D. virescens* isolates identified in southern Poland may pose a greater threat to native European beech than to foreign sugar maple.

## 1. Introduction

*Davidsoniella* is a recently introduced new genus, which is a result of a revision in the family Ceratocystidaceae [1]. There are currently four species included in this genus, and *Davidsoniella virescens* (R.W. Davidson) Z.W. de Beer, T.A. Duong & M.J. Wingf. is designated as a type species [1,2]. Taxonomic affiliation and nomenclature of this species have undergone numerous changes [3]. It was originally described in the United States on hardwood logs and lumber as *Endoconidiophora virescens* by Davidson [4]. The author observed that this fungus possesses growth and morphological features that are very similar to those of *Endoconidiophora coerulescens* Münch, which was described much earlier in Europe [5]. However, he also pointed out some morphological and ecological differences between the two species [4]. Subsequently, both of these species were transferred to *Ceratocystis*, respectively, as *C. virescens* (R.W. Davidson) C. Moreau and *C. coerulescens* (Münch) Bakshi [6,7,8]. However, Hunt [9], in his monographic study, assigned *C. virescens* as synonym, under the name *C. coerulescens*. On the other hand, the author noted that *C. coerulescens* is an exceedingly variable species and recommended that the population on hardwoods should be treated separately, at least as a variety or form [9].

In three subsequent mycological monographs *C. virescens* has not been listed as a separate taxon, but considered synonymous to *C. coerulescens* [10,11,12]. However, further research began to provide evidence of differences between these species and the need to treat them as separate species. Re-examination of Davidson’s authentic material indicated that *C. coerulescens* and *C. virescens* manifest morphological differences in their conidial states [13,14]. Each of these fungal species produces different volatile metabolites, which is reflected in the different odor of cultures [15,16,17]. Isozyme analysis and molecular comparison also clearly separate these two species of fungi [18,19,20,21]. While *C. virescens* was transferred to *Davidsoniella* in the last taxonomic revision, the fungus *C. coerulescens* was placed in the genus *Endoconidiophora* [1].

Significant differences between both species of fungi also concern the distribution, the host plant spectrum and the ecological role in forest stands. *E. coerulescens* occurs in Europe, North America, and Asia, where it causes sapstain on many conifers [5,7,11,22]. *D. virescens* has so far been recorded only in the eastern areas of North America and in Canada [4,10,23,24]. It is reported mainly as a saprotroph on the logs of various hardwood species [4,25,26]. *D. virescens* also occurs as a pathogen, which causes sapstreak disease on *Acer saccharum* Marsh., and in scattered locations on *Liriodendron tulipifera* L. [23,26,27,28,29]. *D. virescens* has also been found sporadically on *A. rubrum* L., *A. saccharinum* L., and *Betula alleghaniensis* Britton, but the disease in this species is little known [24,25,30]. The most characteristic symptom of sapstreak disease is the distinctive stain of internal wood in the roots and at the base of the stem. Stained wood is greenish yellow with reddish streaks and appears watersoaked. In cross-section, the stain streaks have a radial arrangement and are surrounded by a dark-green zone [23,25,27]. The pH of *A. saccharum* wood is normally around 5.5, while wood in the watersoaked areas had pH of 8.5 or higher [27]. The stain significantly lowers the commercial value of affected lumber [23,31]. These symptoms inside the trunk are accompanied by the development of small leaves, as well as gradual branch dieback. In case of symptom progression, this lead to death of the tree within a few years [23,27,29,32,33,34].

*D. virescens* is able to survive and sporulate on air-dried boards for several months [23]. This indicates that it is possible to introduce the fungus together with wood or wood products. There are so far no reports of *D. virescens* from regions outside North America [24,26,35]. *A. saccharum* does not occur in European forests, but there are other maple species, *A. campestre* L., *A. platanoides* L. and *A. pseudoplatanus* L., that could be potentially affected by *D. virescens*. However, there is no convincing evidence that *D. virescens* poses a real threat to *Acer* species other than *A. saccharum* [24]. This results in different assessments of phytosanitary risk and quarantine regulations [30,36]. *D. virescens* has not been considered to be a quarantine pest by the European and Mediterranean Plant Protection Organization [35]. However, this species is listed in Annex II as a Union quarantine pest not known to occur in the EU territory [24]. Import restrictions into the EU mainly apply to wood of *A. saccharum* from Canada and the United States, but there are also other restrictions, and various phytosanitary certificates are required [24]. In this context, it should be noted that, so far, it is not certain whether the saprotrophic forms occurring on various hardwoods in the USA are the same species as the causal factor of sapstreak diseases on sugar maple trees [24]. A draft nuclear genome sequence was developed for the *D. virescens* isolate originated from *A. saccharum*, which should allow for some comparative studies with other isolates [2].

Recently, during research in southern Poland, an accumulation of mycelium and spore-forming structures of the fungus of the *Davidsoniella* genus was found on the cross sections of logs and branches of *Fagus sylvatica*. Research was carried out in order to (i) characterize this fungus in vivo and in vitro, and its identification based on morphology and phylogenetic analysis, (ii) determine the growth rate of its colonies depending on temperature, and (iii) determine pathogenicity towards two plant species, *F. sylvatica* and *A. saccharum*. On this basis, an attempt was made to assess whether *D. virescens* may pose a threat for European forests.

## 2. Materials and Methods

### 2.1. Study Material, Isolation and Microscopic Analyses

Study material was collected on two forest sites (A, B) in southern Poland, approximately 200 km apart (Figure 1). Site A was located in Ojców National Park (N 50°12′16″ E 19°48′48″), and site B in Rozpucie (Brzozów Forest District; N 49°35′24″ E 22°25′9″) (Figure 1). Study material consisted of fragments of logs and branches of *Fagus sylvatica*, on which the presence of blackish patches had been detected on cross sections as an effect of accumulation of mycelium and fungal sporulation structures (hereafter referred to as ‘blackish fungal growth’—BFG).

Beech material on site A came from an old beech tree (over 90 years old, 75 cm in diameter at the base of the trunk), consisting of two trunks. One of them was broken under the influence of the wind at a height of 1.6 m (Figure 2a). Due to the forest road being blocked, the trunk was cut into several fragments, which were left on the side of the road. Beech material on site B remained as a result of previously carried out cuts of individual trees. On site A, 12 small wood slices together with fungal structures were sampled from 12 different places on cross sections of logs in August and October 2018 (Table 1). From site B, 12 fragments of logs and branch bases (length 4–15 cm, diameter 7–18 cm) were collected in August 2021 (Table 1). The samples were placed separately in plastic bags and brought to the laboratory, where they were stored in a cold room at 5 °C and subjected to microscopic analysis.

From each sample, 4–6 slides were prepared, and individual elements constituting the BFG were measured. The aim was to initially identify the fungus species based on morphological features and to determine whether the BFG in all samples was created by the same fungus. Appropriate mycological keys and monographs were used for identification [1,4,9,10,11,12,13].

In vitro experiments were performed on malt extract agar (MEA: 20 gL^−1^ Biocorp malt extract, 20 gL^−1^ Biocorp agar (Biocorp Polska Sp. z.o.o., Warsaw, Poland), supplemented with 100 mL^−1^ streptomycin sulphate (Polfa S.A., Warsaw, Poland)) in Petri dishes (diam. 9 cm). Fungal isolations were performed within 24–72 h after samples had been collected from the forest sites. Samples were sectioned. The exposed surface was disinfected by washing with cotton wool soaked in 96% ethanol. Then, small fragments of wood were taken using a sterilized scalpel after removal of superficial tissue. From each sample, 6–12 pieces of wood were taken. From the samples from site B, which showed a grey-brown discoloration of the wood, 12–24 fragments were additionally taken for isolation, 3–12 cm below the BFG. These fragments were placed in Petri dishes on the surface of malt extract agar. The incubation took place at 20 °C in darkness. All Petri dishes were examined every 3–5 days for at least 3 weeks. Cultures showing morphological structures characteristic of BFG in vivo were isolated on MEA in new Petri dishes. One representative culture was selected for each sample. Then, their morphological and molecular characteristics were examined. Individual structures were mounted in distilled water, and 20 to 60 measurements at 400× or 1000× magnification were made of each structure; the extreme values and standard deviation (SD) are given. Observations were made with a Zeiss V12 Discovery stereomicroscope and with a Zeiss Axiophot light microscope (Zeiss, Göttingen, Germany), using differential interference contrast or phase contrast illumination. The specimens examined are preserved in the Department of Forest Ecosystems Protection, University of Agriculture in Kraków, Poland.

### 2.2. DNA Extraction, PCR, Sequencing and Phylogenetic Analyses

To verify the morphology-based identification, the nucleotide sequences for five gene fragments of representative cultures were determined: 18S; the internal transcribed spacer regions ITS1 and ITS2, including the 5.8S gene (ITS); 28S region of the ribosomal RNA (rRNA), β-tubulin 2 (TUB2) and translation elongation factor 1-α (TEF1). Genomic DNA extraction, polymerase chain reaction (PCR) amplification and sequencing reactions of the isolates were carried out according to the procedure described in detail by Kowalski et al. [37]. The 18S, ITS, 28S regions and the TUB2 and TEF1 genes were amplified using the primers [38,39,40,41,42,43] listed in Appendix A (Appendix A). Due to the difficulty in obtaining gene fragment sequences of TUB2 and TEF1 for the isolates studied, new primers were designed and used in PCR. For TUB2, it was the forward primer T10CerPB 5′-TCTATAGGTCCACCTCCAGAC-3′, and for TEF1, it was the reverse primer EFCF6PB 5′-CATRTCACGGACGGCGAARC-3′. The latter oligonucleotide was obtained by modifying the EFCF6 primer proposed by [43].

The resulting 18S, ITS and 28S sequences were concatenated because they contained overlapping fragments. Two adjacent fragments of the TEF1 gene were treated similarly. All obtained gene fragment sequences in this study were deposited in GenBank with the accession numbers shown in Table 1. The sequences obtained from representative cultures were used as queries in searches using the BLASTn algorithm [44] to retrieve sequences of taxa closely related to them from GenBank (http://www.ncbi.nlm.nih.gov, accessed on 7 May 2024) for phylogenetic analysis. Individual datasets for the five gene fragment regions were used for phylogenetic analyses using methods following Kowalski et al. [37]. The datasets contained all species of the genus *Davidsoniella* for which at least one of the gene sequences was available in the GenBank, with *Endoconidiophora coerulescens* and *E. polonica* included as an outgroup (Appendix A) following de Beer et al. [1]. The best evolutionary substitution model for dataset 18S was HKY+I; for ITS, it was GTR+I; for 28S, it was HKY+G; and for the protein-coding gene datasets TUB2 or TEF1, it was GTR+G and GTR+I+G, respectively. The resulting alignments and trees were deposited into TreeBASE (http://purl.org/phylo/treebase/phylows/study/TB2:S31400, accessed on 19 May 2024).

### 2.3. Colony Growth in Relation to Temperature

A temperature assay was performed in vitro on 2% MEA in Petri dishes. Six *D. virescens* isolates obtained from *F. sylvatica* wood in southern Poland were used for the test (Table 1). Mycelial plugs (8 mm in diameter) from the edge of actively growing 5-day-old colonies were transferred into the center of dishes and incubated at 5, 10, 15, 20, 25, 30 and 35 °C in the dark. Two replicated plates were prepared for each fungal isolate and temperature. Colony diameters were measured after 8 days with a millimeter ruler in directions perpendicular to each other. The average diameter from measurements in each replicate was calculated.

### 2.4. Pathogenicity Test

*Fagus sylvatica* and *Acer saccharum* seedlings were used for the pathogenicity test. Two-year-old seedlings of *F. sylvatica* were obtained in autumn 2021 from a forest nursery in the Daleszyce Forest District (N 50°47′39″ E 20°42′36″). They were immediately planted in sandy clay, moderately moist soil (30 × 30 cm spacing), in a fenced area on the outskirts of Kraków, where there was side shade from neighboring low trees and shrubs. In 2022, all individuals were developing well, without any disease symptoms. Dimensions of seedlings at the time of inoculation were as follows: height 0.5–0.8 m, diameter at the stem base 0.5–0.7 cm.

Two-year-old seedlings of *A. saccharum* were obtained in June 2022 at the ornamental tree nursery in Radomyśl Wielki near Mielec (eastern Poland). They were grown individually in pots (height 15 cm, diameter 18 cm) on a substrate composed of peat and perlite. They were located next to a plot with *F. sylvatica*. At the time of inoculation, their dimensions were height of 0.5–0.9 m and diameter at the stem base of 0.5–8 cm. Both plant species were regularly observed and watered as required.

Inoculation was carried out in the first days of August 2022 in ca. 7 mm long wounds made with a sterile scalpel on stem approximately 15 cm above the root collar. Six *D. virescens* isolates obtained from *F. sylvatica* wood in southern Poland were used for the test (Table 1). Six *F. sylvatica* and six *A. saccharum* seedlings were inoculated with each isolate. Inoculum production, inoculation and re-isolation were carried out according to the methods used by Kowalski et al. [45]. Briefly, isolates were grown on MEA for 5 days at 20 °C in darkness. Small sterile beech-wood sticks (ca. 5 × 2 × 2 mm) were then placed on the colonies and incubated further for 2 weeks. These beech-wood pieces used for inoculation were overgrown by mycelium along with endoconidia and ascospores and the structures producing them (phialophores, perithecia). For control inoculations, sterile beech-wood sticks were kept on fungus-free MEA for 2 weeks. In total, 72 stems were inoculated. Control inoculations were carried out with sterile wood pieces applied to wounds of six stems of each plant species. The symptoms were examined 4 months after inoculation. The lengths of superficially visible necrotic lesions were measured, and the occurrence of fungal fructification was recorded. After removal of the bark and superficial layers of wood, the extent of wood discoloration was determined. If a given seedling began to die earlier, the analysis was performed immediately after observing symptoms in the crown. Re-isolations were attempted from all inoculated and all control stems within 24 h from harvesting. Three to eight (over 1400 in total) tissue pieces were taken from inoculation wounds, the lesions and discolored wood. The isolation was considered positive for a given seedling if the tested fungus grew from at least one tissue sample.

### 2.5. Statistical Analyses 

Differences between groups within the study variables were given an analysis of variance (ANOVA) after log_10_(x + 1) transformation of the data and checking that they met the assumptions of homogeneity of variance. If the data did not meet these assumptions, a post hoc test of multiple comparisons of the mean ranks for all groups of variables following Kruskal–Walis test was applied. For ANOVA, the Tukey HSD post hoc test was performed. All statistical analyses were performed using the program Statistica 13.1 [46].

## 3. Results

### 3.1. Symptoms on Fagus sylvatica Logs In Situ

*D. virescens* was first observed in August 2018 during research in Ojców National Park (site A) (Figure 1, Table 1). On a trunk of a broken beech (on the lying and standing parts), blackish fungal growths (BFGs) were found on the surface of bark-free wood (Figure 2a). In the remaining cases, they were formed on cross-sectional surfaces of the logs, 13 to 27 cm in diameter (Figure 2b,c). BFGs were of different sizes (Figure 2b,c); in only one log, fungal growth occupied the entire front surface. Observation in October 2018 showed that the blackish structures were covered with a fine white powder (Figure 2b).

Very similar symptoms were observed in August 2021 in Rozpucie (Brzozów Forest District, site B, Figure 1). They occurred on 12 log fragments and the bases of branches lying on the ground or in the heap. They always occurred on exposed wood, never on bark. They took the form of either uniform, quite extensive structures (Figure 2d,e) or numerous separate patches sometimes merging with each other (Figure 2f). In five logs, BFG covered over 75% of the cross-sectional area (Figure 2e). In some logs, BFG was covered with a fine white powder (Figure 2e). The wood under the BFG had grey-brown discoloration to a depth of 12 cm (Figure 2g–i). The shape of the wood discoloration corresponded to a large extent to the shape of BFG on the front face of the logs (Figure 2f–i). Of all analyzed logs, *D. virescens* cultures were obtained from 17 samples (Table 1).

### 3.2. Morphological Characterization of Davidsoniella virescens

#### 3.2.1. In Vitro

Colonies on MEA were growing rapidly; after 8 days at 20 °C, they reached 5.5–8.3 cm in diameter. Mycelium was dark greenish grey, initially quite fluffy (Figure 3a), later suppressed to the medium (Figure 3b). Superficial hyphae were subhyaline to greenish grey, smooth, septate, branched, 2.5 to 6.0 µm thick. Immersed hyphae were dark grey, smooth, 4–8 µm thick. The colony was covered with a white fine powder in places due to the production of asexual spores. Young colonies emitted an intense, sweet odor, which became less pleasant in older colonies.

The colonies produced an abundant asexual stage, characteristic of the *Chalara* genus (Figure 4a–n). Two types of phialophores developed, differing in frequency, shape, length and width dimensions. Phialophores of the first type (type A) were simple, only occasionally with a branch at the base, cylindrical, olivaceous brown and 3–9 (14) septate, 42–325 µm (169.3 ± 74.7 µm; mean ± SD) long, and 4.5–6.2 µm wide at the base, terminating in a phialide (Figure 4a,b). The phialides were subcylindrical, 30–79 µm long, with hyalin, subcylindrical to conical, 2.5–3.7 µm wide collarette (Figure 4a–c). Endogenously formed conidia were often arranged in long chains. They were unicellular, hyaline, smooth, thin walled, cylindrical, initially with truncate, later with rounded ends, 5.0–17.5 µm (9.8 ± 2.8 µm) long and 2.0–5.0 µm (3.0 ± 0.8 µm) wide, often with polarly located small oil droplets (Figure 4c,f–h).

Phialophores of the second type (type B) were olivaceous brown, 3–10 septate, 60–225 µm (121.9 ± 43.0 µm) long, 5–7 µm wide at the base, and slightly enlarged to 6.2–7.5(–9.0) µm at the top (Figure 4d,e). Endoconidia were hyaline, unicellular, oval to barrel-shaped, and globose, 5–13 × 4.5–13.0 µm (8.1 ± 2.0 × 6.8 ± 1.8 µm), occurring singly or in long chains (Figure 4i,j). After germination of type B endoconidia, the germ hypha can transform into a type A phialide that produces type A endoconidia (Figure 4k). Phialophores of type A predominate quantitatively in young cultures (Figure 4a). Phialophores of type B are rare in young cultures, but quite common in older ones (Figure 4d,e).

In the cultures, after 3–5 days, on superficial mycelium or partly embedded in agar, perithecia began to be produced abundantly (Figure 3a), and over time appeared densely throughout the colony (Figure 3b). Perithecia were black, flask shaped to globose; 120–250 µm (181.2 ± 33.1 µm) high, 125–250 µm (180.5 ± 32.0 µm) wide; ornamented with numerous dark, septate, smooth hyphal hairs; 36–350 µm (179.3 ± 72.8 µm) long, and 4.0–5.0 µm wide (Figure 4l). Necks were black, becoming lighter towards the apex, straight or slightly curved, 480–1300 µm (801.7 ± 183.0 µm) long excluding ostiolar hyphae, 35–50 µm wide at the base, and 10–12(–15) µm wide at the top. Ostiolar hyphae were divergent, hyaline, 16–30 µm long, 1.2–1.5 µm wide at the base, tapered slightly to blunted tips (Figure 4l). Ascospores were collecting in sticky white spherical masses at the top of neck. Ascospores were unicellular, hyaline, cylindrical to oblong, straight to slightly curved with polar oil droplets, surrounded by transparent sheath, which appeared uniform or slightly thicker at the spore ends; 5.0–9.0 × 1.5–2.5 µm (7.0 ± 0.8 × 2.0 ± 0.2) without sheath and 7.0–11.0 × 2.0–4.0 µm (8.8 ± 0.8 × 2.7 ± 0.4) µm with sheath (Figure 4m,n).

#### 3.2.2. In Situ

Each structure observed in vitro was produced by *D. virescens* in situ on *F. sylvatica* wood. They were similar in shape and size, hence only their main characteristics are given below. Perithecia formed on the surface of the wood or were slightly embedded in the substrate; brown-black, flask shaped, spherical or slightly flattened at the base; 107–200 µm (153.2 ± 21.0 µm) high, 120–180 µm (153.1 ± 17.8 µm) wide; ornamented with numerous dark, septate, hyphal hairs, 30–300 µm long (157.6 ± 67.9 µm) and 3.0–5.0 µm wide. Necks were black, becoming lighter toward the apex, 450–1100 µm (683.0 ± 181.7 µm) long excluding ostiolar hyphae, 27.5–42.5 µm at the base, 10–15(–22) µm wide at the top. Ostiolar hyphae were hyaline, divergent, 15–42.5 µm long. Ascospores were unicellular, hyaline, cylindrical to oblong, straight or slightly curved with oil droplets, 5.5–8.0 × 1.8–2.2 µm without sheath and 7.0–9.0 × 2.0–2.7 µm with transparent sheath.

Phialophores of type A were simple, occasionally with one branch at the base, olivaceous brown, 3–9 septate, 52–325 µm (172.4 ± 58.5 µm) long, (3.2–)5–7.5 µm wide at the base, terminating in a phialide. The phialides were hyalin to light olive, subcylindrical to obclavate, 55–77 µm long. Collarettes were 2.0–3.7 µm wide. Endogenously produced conidia (type A) were unicellular, hyaline, thin walled, cylindrical with truncate or rounded ends, 5–13 × 1.8–4.0 µm (8.0 ± 2.1 × 2.3 ± 0.4 µm), with small oil droplets, often arranged in long chains. Phialophores of type B were olivaceous brown, 3–5 septate, 45–125 µm long, 3.7–6.0 µm wide at the base, 5.0–7.0(–9.0) µm wide at the top. Endoconidia (type B) were hyaline, unicellular, oval to barrel-shaped, or spherical, 5.0–9.0 × 4.0–9.0 µm (7.6 ± 0.9 × 6.5 ± 1.3 µm), occurring singly or in short chains.

On sites A and B, it was observed that some blackish structures were entrained with a fine white powder. Microscopic analysis showed that this was the result of the formation of type A and B endoconidia. Moreover, in some samples, quite numerous conidiophores and conidia of another fungus of the *Clonostachys* genus were developed.

### 3.3. DNA Sequence Data and Phylogenetic Analysis

Alignments for the 18S, ITS, 28S, TUB2 and TEF1 sequences respectively contained 1736, 520, 825, 1106, and 1563 characters (including gaps). The datasets used in the phylogenetic analysis contained different numbers of variable characters: 18S—5, ITS—59, 28S—20; TUB2—298 and TEF1—273, of which 3, 51, 13, 108 and 219 were parsimoniously informative, respectively. The aligned TUB2 gene region consisted of introns 1–5 with exons 2–6. The exon/intron arrangement of the TEF1 data included exons 3, 4 and 5 interrupted with introns 3 and 4. The results of phylogenetic analysis of the nucleotide sequences of five gene fragments of isolates from Poland indicated that they were most similar phylogenetically to *D. virescens* isolates from the USA. The isolates of this species formed a distinct clade with strong statistical support on all phylogenetic trees except the tree obtained from the analysis of the 18S gene fragment (Figure 5, Figure 6 and Appendix A). TEF1 was the most phylogenetically robust among all markers tested (Figure 6). The topology of the trees obtained for each of the three genes, i.e., 28S (Appendix A), TUB2 and TEF1 (Figure 5 and Figure 6), within the *D. virescens* clade indicated the existence of two distinct phylogenetic lineages associated with different host plants. Distinguishing at least one of these in each of the three analyses was statistically significant. The first group of isolates was derived from trees of the genus *Fagus* or *Quercus*, while the second was associated with *A. saccharum*. All Polish isolates obtained from *F. sylvatica* showed affinity with North American isolates derived from *F. grandifolia* and formed a distinct phylogenetic lineage with them. Phylogenetic analysis of protein-coding gene sequences (TUB2, TEF1) indicated the presence of intraspecific variability among *D. virescens* isolates. Polish isolates were characterized by relatively low variability in these genes, in contrast to North American isolates, especially with regard to TEF1 sequences (Figure 5 and Figure 6).

### 3.4. Colony Growth in Relation to Temperature

Colonies of all six tested isolates of *D. virescens* showed growth on MEA in the temperature range of 5 to 30 °C, and occasionally also at 35 °C (Figure 7). The optimal temperature for radial colony growth was 20 °C. However, at 25 °C, colony growth was only slightly slower. The differences between colony diameter at this temperature compared to the optimal temperature were not statistically significant (Figure 7). Above 25 °C, colony growth was significantly slower, and the differences between colony diameters growing at 25, 30 and 35 °C were statistically significant, according to the Kruskal–Wallis test (Figure 7).

### 3.5. Pathogenicity Test

#### 3.5.1. *Fagus sylvatica*

Out of 36 seedlings of *F. sylvatica*, inoculated in stem wounds with *D. virescens*, five seedlings showed wilting symptoms throughout the entire crown after 3–6 weeks. The leaves dried quite suddenly, turned pale green or greyish-green, less often brown, and wrinkled or curled at the edges (Figure 8a–p). Symptoms in the crown were accompanied by bark necrotic lesions and wood discoloration within the stems, occasionally in the roots (Figure 8a–p). Synthetic descriptions of symptoms in these seedlings are given in Table 2.

Among the remaining 31 seedlings, no symptoms of wilting or early leaf discoloration were observed. Natural discoloration of the leaves occurred only due to the arrival of autumn. During the evaluation 4 months after the inoculation, 11 seedlings had bark necrosis around the inoculation site with a length of 1.3 to 7.2 cm (Table 3). In 54.8% of the stems, the wounds were not healed; in the remaining cases, the wounds were partially healed. In 6.5% of stems in the wounded area, *D. virescens* produced perithecia. The most noticeable internal symptom was the discoloration of the wood (Figure 9a–l). On six stems, their length did not exceed 5 cm (Figure 9a–c). In the remaining stems, the wood was discolored over a length of 5.1 to 15.6 cm (Figure 9d–i, Table 3). On most stems within the wound inoculations, the wood was dark grey, while below and above this area, the wood was light brown or grey-brown (Figure 9a–i). Sometimes this discoloration was not uniform, but there were darker longitudinal streaks (Figure 9h,i). The longitudinal section of some stems clearly showed that wood discoloration was associated with bark necrosis (Figure 9h,i, Table 3). A hand-drawn tangential section of discolored wood showed few hyphae of *D. virescens* in the vessels and in ray parenchyma (Figure 9j,k).

#### 3.5.2. *Acer saccharum*

None of the 36 inoculated seedlings showed any symptoms of early leaf discoloration or wilting, as well as bark necrosis around the wound. In 47.2% of the stems, the wounds were not healed, and in the remaining cases, the wounds were partially healed. Perithecia of *D. virescens* developed on two (5.6%) stems in the wound area. There was discoloration of the wood inside all stems, starting from the inoculation site. In two seedlings, the wood was discolored brown (Figure 10a); in the remaining ones, the wood was dark grey with an olive green tint (Figure 10b–j). The length of the discolored wood ranged from 1.2 to 6.1 cm (Table 3). In 20 (55.6%) seedlings, the length of the discolored wood did not exceed 2 cm (Figure 10b–d). In some stems, the discoloration was irregular, with dark streaks extending both upwards and downwards (Figure 10e–h). Sometimes they formed quite thin lines (Figure 10i,j). A hand-drawn tangential section of discolored wood showed few hyphae of *D. virescens* in the vessels (Figure 10k).

All *D. virescens* isolates caused greater wood discoloration and bark necrotic lesions in *F. sylvatica* than in *A. saccharum* (Figure 11, Table 3). Most of the differences found in the extent of discoloration between host plants were statistically significant (Table 3). The discoloration caused by all *D. virescens* isolates in *F. sylvatica* was significantly greater than in the control. However, none of the isolates tested on *A. saccharum* caused significantly greater discoloration compared to the control (Table 3).

#### 3.5.3. Re-Isolation and Control

*D. virescens* was re-isolated from all five seedlings of *F. sylvatica* showing wilt symptoms. Its colonies grew from a total of 40.6% of wood fragments (Table 4). Some differences were found in the frequency of its isolation depending on the place where wood fragments had been collected (Table 4). In the case of seedling no. 3/Fs509K, *D. virescens* was also re-isolated from discolored wood in roots. Of the 31 *F. sylvatica* seedlings without wilt symptoms, a positive re-isolation result was obtained in 26 of them (Table 4). *D. virescens* colonies grew from 19.0% of wood fragments (Table 4). From 36 inoculated stems of *A. saccharum*, 584 discolored wood fragments were placed on MEA. *D. virescens* was isolated from only 1.0% of such fragments from five (13.9%) seedlings (Table 4). Colonies of fungi of the genera *Alternaria*, *Aureobasidium, Cadophora*, *Cladosporium*, *Clonostachys*, *Colletotrichum*, *Epicoccum*, *Fusarium*, *Paraconiothyrium*, *Pezicula*, *Phoma*, *Phomopsis*, and *Trichoderma* were also isolated from wood fragments of both plant species. Among the *F. sylvatica* isolates, *Clonostachys* sp. predominated, while *Fusarium* spp. were the most frequently isolated colonies from *A. saccharum*.

No control seedlings of *F. sylvatica* and *A. saccharum* developed bark necrosis on stems. In the area of the wound in *F. sylvatica*, the wood became light brown in color at a length of 1.1–2.2 cm (Figure 9l, Table 3). In *A. saccharum*, the length of discolored wood was 0.9–1.8 cm (Figure 10l, Table 3). The wood was uniformly brown or dark grey with an olive tint (Figure 10l). *D. virescens* was not isolated from any control seedlings (Table 4).

## 4. Discussion

*Davidsoniella virescens* has so far been found only in North America [4,26,47]. The discovery of this species in Poland is the first report from outside the previously known area of occurrence. *Fagus sylvatica* has not yet been mentioned among the colonized spectrum of trees [4,26,47]. In some areas of the United States, *D. virescens* inoculum is readily available. The fungus often produces mycelium along with sporulating structures on recently cut surfaces of stumps and logs [4,23,27,34,48,49]. Current observations in Poland at two significantly distant forest sites led to similar conclusions. Mycelium and sporulation structures of *D. virescens* did not form on the bark of the currently analyzed logs. The same was observed in inoculated plant seedlings, in which perithecia developed only occasionally (11.1% of beech, 5.6% of sugar maple) at the site of injury. Regardless of whether colonization occurred naturally (logs) or as a result of artificial inoculation (seedlings), the wood showed symptoms of grey or grey-brown discoloration, which was uniform or marked with dark streaks.

Colonized wood plays a significant role in the development of *D. virescens*. First, spores formed very abundantly on the wood surface can be easily spread throughout the tree stand. Conidia, due to the lack of mucus, might be spread by the wind. However, ascospores are surrounded by mucus, so rain and insects are more likely involved factors. The great importance of insects in the spread of pathogenic and saprotrophic Ceratocystidaceae fungi is well documented [1,12,50,51,52]. Such suggestions also apply to *D. virescens* [26,29,53]. Secondly, on logs and stumps, where we observed sporulation, the fungus was most likely able to survive the winter, because its mycelium was growing deep into the wood, which was confirmed by isolation on agar medium. For a fungus that does not produce sclerotia, this is an important way to survive unfavorable external conditions.

### 4.1. Morphological, Phylogenetic and Physiological Aspects

So far, the morphological features of *D. virescens* have been given in detail when a new species is described (as *Endoconidiophora virescens*) [4]. These features were also partially described during the re-examination of authentic Davidson’s strain [13,14]. Davidson [4] indicated that a characteristic feature of *D. virescens* is the production of two types of phialophores, narrow (2.2–3.5 µm) and wide (5–6.5 µm), and the corresponding two types of endoconidia. The microscopic characteristics of the isolates now obtained from *F. sylvatica* are largely consistent with those described for *D. virescens* by Davidson [4]. However, some differences can be noticed, especially regarding the *Chalara* asexual stage. Due to the diversity of phialophores and the endoconidia they produce, for some systematization, the following terms have been introduced for them: ‘type A’ and ‘type B’. In addition, greater variation in the size and shape of type A and B endoconidia has now been found. In the isolates currently analyzed, type A endoconidia were 2–4(–5) µm wide, whereas according to Davidson [4] they were 2–3 µm wide. Davidson [4] reports that endoconidia of the second type (=type B) are short barrel-shaped with dimensions of 5–9 × 5–6.5 µm. Currently, in addition to such spores, also globose spores, even over 9 µm in diameter, have been found both in situ and in vitro. Present observations indicate that changes in size and shape occur over time after the endoconidia leave the phialides. Most likely, Davidson [4] gave the dimensions of the endoconidia when they were still inside the phialides or shortly after they left them. However, without a direct comparison of the current cultures with fresh cultures from *A. saccharum*, it is not possible to determine whether this interpretation is justified. If *D. virescens* colonizing *A. saccharum* did not form globose conidia under any conditions, this would be a significant feature distinguishing it from the fungus colonizing *F. sylvatica*. The increased diversity of asexual forms in *D. virescens* is also influenced by the production of endoconidia by germinating type B endoconidia. Such a phenomenon has not been reported in this fungus so far. High variability in the size of conidia in cultures of *D. virescens* derived from *A. saccharum* was pointed out by Richter [29]. However, this author did not provide a detailed description of either conidia or phialophores. It was only reported that conidiospores (approximately 6–25 × 3–6 μm in size) were formed directly from hyphae. According to this author, ascospores reached a length of 6–7 μm in vitro, while in the current cultures, they were 5–9 μm long (without gelatinous sheaths).

In the present work, a comprehensive phylogenetic analysis of fungi of the genus *Davidsoniella* was performed with the use of multiple fragment sequences of five selected genes. This study has shown that four of them, i.e., ITS, 28S, TUB2 and TEF1, are highly useful for the identification of species belonging to this genus. Phylogenetic analysis of three (28S, TUB2, TEF1) of the five gene fragments tested indicated the existence of two lines of *D. virescens* isolates dependent on host plants. This phenomenon was first pointed out by Harrington et al. [18] on the basis of a study of the variability of *D. virescens* isolates using 10 isozymes. According to another study by Harrington et al. [54], much of the variation in *D. virescens* nuclear markers appears to be due to differences between two forms of the species, one associated with *Acer* and *Liriodendron* and the second associated with *Fagus* and other hardwoods. These results were later confirmed by Witthuhn et al. [20], who performed a phylogenetic analysis of selected fungal species, including *D. virescens*, based on MAT-2 DNA sequences of isolates obtained from different host plants. Differences in TEF1 gene sequences between *D. virescens* isolates obtained from *Acer* and *Fagus* were also evident in the phylogenetic tree presented by Harrington [55]. Unfortunately, this author used single *D. virescens* sequences representing each lineage, which made it impossible to determine whether these groups differed statistically. In the present study, the identified two groups of isolates received high statistical support according to the results of each of the three phylogenetic analysis methods used (ML, MP and BI). According to Witthuhn et al. [20], since *D. virescens* isolates fall into two groups based on phylogenetic analysis and host plant specificity, they “should be recognised as distinct species”. However, resolution of this question requires further detailed research.

Numerous Ceratocystidaceae produce intense odor. It is assumed that this is an adaptation for fungal dispersal by insects [1,4,9,12,17,56,57,58]. *D. virescens* develops a distinctive musty odor on lumber and in culture, while a similar species, *E. coerulescens*, produces a banana-oil odor [4]. According to other authors, a colony of *D. virescens* obtained from sugar maple tree had a sweet, musty odor [27,28,29]. In the cultures of *D. virescens* from *F. sylvatica*, a sweet but not musty odor was noticeable. This requires further explanation because the production of some metabolites may be influenced by strain-dependent feature and culture conditions [17]. Birkinshaw and Morgan [15] identified the products which are responsible for the odors in *D. virescens* and *E. coerulescens* cultures. The first one produced the same methylheptenone and mixture of L- and DL-methylheptenol as *E. coerulescens*, but in different proportions. However, *D. virescens* did not produce isobutyl acetate. As in the case of *Ceratocystis platani* (Walter) Engelbr. & T.C. Harr., volatile organic compounds may be useful as biomarkers for comparison of various strains of *D. virescens* from Europe and North America [58].

One of the basic factors influencing the development of fungi and the establishment of alien fungi in new areas is temperature [50,59,60,61]. Temperature may also influence the rate at which vascular pathogenic fungi cause disease symptoms [62,63]. Current tests have shown that *D. virescens* colonies originating from *F. sylvatica* are able to grow in a wide temperature range, from 5 to 30 °C, and some isolates even up to 35 °C. This means that they can colonize wood in Poland at different times of the year. It is sometimes observed that *D. virescens* colonies show a variable growth rate in vitro at the same temperature. One of the reasons for this phenomenon may be the high sensitivity of *D. virescens* colonies to prolonged storage on agar media in vitro. They may lose not only the ability of rapid radial growth but also the ability to produce perithecia [4]. Webber [26] analyzed the temperature range in the distribution zones of *D. virescens* on *A. saccharum* in North America and concluded that this fungus would find suitable conditions in the event of an accidental introduction in Great Britain. This also applies to the possibility of development in the areas where the native *Acer* species occur in continental Europe [47]. 

### 4.2. Pathogenicity of D. virescens Isolates from F. sylvatica 

Pathogenicity is the ability of a given fungus to cause disease, while virulence determines the degree of pathogenicity [64]. The life style of *D. virescens* originating from southern Poland is unknown. The current tests were primarily aimed at determining whether it is capable of causing disease symptoms in living plants. The factor that often differentiates the pathogenicity of fungi is the plant host [64]. Two plant species were used in the current tests, *F. sylvatica* and *A. saccharum*. The pathogenicity test was conducted by placing fungal inoculum in wounds formed on the stems of both tree species. This should be a favorable circumstance for *D. virescens*, because wounds play an important role in the process of infection of trees by this fungus. In North American forest stands, *D. virescens* infects *A. saccharum* usually through wounds at the root collar and at the base of the stems, which are formed during human activities such as forest thinning, logging, road building, or sap hauling [23,27,29,32,34,49,65].

Wood discoloration was observed in all currently inoculated seedlings, and its extension was very diverse. It should be taken into consideration that as a result of wounding living sapwood, even without the participation of fungi, wood discoloration occurs as a result of compartmentalization [27,66,67,68,69]. This could currently be observed in control plants. In case of infection by weak fungal pathogens through wounds, colonization is typically limited to a small area around the wound site [51]. Numerous pathogenic species from the Ceratocystidaceae may extensively colonize sapwood tissue and cause discoloration or staining as well as wilting and death of whole trees [51,70]. Some of them, e.g., *C. platani* on infected *Platanus* trees, after axial expansion in sapwood, can travel through the medullary rays back to the bark, causing its necrosis at various heights of the stem. That results in canker stain, a disease leading to mortality of plane trees [70,71]. However, other fungi, e.g., *Bretziella fagacearum* (Bretz) Z.W. de Beer et al., spread in the xylem vessel and induce the formation of tyloses and gums, which impair water transport, thus causing symptoms typical of true vascular disease [50,62]. According to Hepting [27], *D. virescens* develops very numerous hyphae in the vessels, and only rarely in the rays. This author suggests that watersoaking occurs as a result of the death of the surrounding ray and wood parenchyma cells. In places where the sapstreak comes into contact with the cambium, necrosis of this tissue occurs and may form elongated cankers [23,27]. Such a process cannot be excluded in some *F. sylvatica* seedlings showing stem bark necrosis. However, in the currently inoculated plant species, only a few hyphae were observed in the vessels.

Symptoms in both plant species were evaluated 4 months after inoculation. It is difficult to predict the further effects of the internal changes in the form of wood discoloration. In addition to wood discoloration, 30.6% of *F. sylvatica* seedlings developed bark necrosis, which would probably lead to their death in the following weeks. The emergence of wilt in the whole crown in five seedlings within 3–6 weeks after inoculation should be considered particularly disturbing. Based on comparison with other dangerous vascular pathogens, this rate of wilting can be considered as very fast [14,72,73]. It is difficult to indicate the factors that had a decisive influence on this. The involvement of certain biochemical compounds in causing wilt symptoms in inoculated *F. sylvatica* seedlings cannot be ruled out. In members of the Ceratocystidaceae, catechol dioxygenase (CDO) enzymes were identified as pathogenicity and virulence factors. These enzymes are involved in the degradation of phenolic plant defense compounds. It was shown that the genomes of most necrotrophic pathogens, including *D. virescens*, contained four different genes encoding CDOs, those of weak pathogens contained two to three genes, and those of saprotrophs had only a single gene [74].

The reason may lie in the high susceptibility of some host plant individuals that were unable to limit the rapid axial spread of the pathogen [75]. *F. sylvatica* is characterized by high genetic, biochemical and physiological diversity, which was the basis for distinguishing two (early and late) phenological forms. Individuals representing these forms differ in the timing of leaf development and autumn fall, frost resistance, level of reactive oxygen species production and profile of antioxidative enzyme activity [76,77,78]. In turn, late frosts are considered as important ecological events that strongly affect the genetic structure of beech, its radial growth, vitality and competitiveness [76,79]. Such aspects are important in the case of tree susceptibility to other pathogens. The differences in susceptibility of *Quercus* species towards *B. fagacearum* has generally been attributed to differences in anatomical features and physiological responses of the host to infection [50]. In the case of Dutch elm disease, trees of *Ulmus minor* Mill. susceptible to *Ophiostoma novo-ulmi* Brasier had significantly wider and longer vessels [80,81].

In the case of one wilting beech seedling, wood discoloration extended to the roots. This phenomenon is noteworthy because some pathogenic fungi, e.g., *C. platani*, *B. fagacearum*, and *Davidsoniella australis*, can move between neighboring trees through functional root anastomosis, which accelerates their spread in forest stands [50,51,70]. Such situations were also observed in the case of *D. virescens* in naturally infected sugar maple stands [27,82]. Development of the sapstreak disease is often more rapid and extensive in roots than in stems [23].

*A. saccharum* is one of the primary deciduous tree species exhibiting sapstreak disease caused by *D. virescens* in North America. Infected trees show generally symptoms in the crown, inside the base of the trunk and in the roots [23,28,35,47,83]. In currently tested seedlings of *A. saccharum*, after 4 months, the average vertical extension of stained wood was 2.08 to 2.90 cm. However, Smith and Houston [75] found mean vertical discoloration at a length of 2.8 cm 7 weeks after inoculation of A. saccharum seedlings with *D. virescens*. In the current experiment, neither bark necrosis on the stem nor disease symptoms in the crown were observed in *A. saccharum*. Houston [23] observed that older sugar maple trees, that died quickly, showed extensive vascular staining, whereas in trees that never developed severe foliar symptoms, discoloration was usually strongly restricted by the tree. In the currently inoculated *A. saccharum* seedlings, discoloration of xylem was quite limited, which could be seen by comparison with *F. sylvatica*. In addition, the study showed that *D. virescens* was significantly less likely to be re-isolated from stained xylem of inoculated *A. saccharum* seedlings than in the case of *F. sylvatica*. However, Hepting [27] regularly re-isolated *D. virescens* from inoculated sugar maple trees if the tissue was grey, while isolations from watersoaked or green-streaked tissue yielded either no organisms or different kinds of bacteria. *D. virescens* may affect the internal wood chemistry. In vitro tests have shown that this species can produce volatiles that enhance the growth of other fungi [2,84]. In the case of canker stain of plane trees, *C. platani* rapidly loses its isolability in vitro if wood has been colonized by saprotrophs [70]).

In North America, it has not yet been clarified whether strains colonizing logs of various hardwood species saprotrophically can cause disease symptoms on living tree species, including sugar maple. Only Harrington et al. [54] reported that an isolate from beech did not cause the disease of *A. saccharum*. However, the pathogenicity of isolates from sugar maple trees with sapstreak symptoms towards trees of the same species has been demonstrated [27,75]. The present results indicate that the *D. virescens* isolates found on *F. sylvatica* in Poland may pose a greater threat to *F. sylvatica* than to *A. saccharum*. This result is quite unexpected. When analyzing the phytosanitary risk from *D. virescens* in Europe, the main consideration was the threat to the native *Acer* species [26,47]. Many aspects related to the appearance of *D. virescens* in Europe require further scientific explanation in order to provide the basis for determining the phytosanitary risk for our forest stands.

## 5. Conclusions

In 2018 and 2021, the fungus *Davidsoniella virescens* was identified on the wood of *Fagus sylvatica* in southern Poland based on morphological and phylogenetic studies. This is the first record of this species outside North America. *F. sylvatica* was demonstrated for the first time as a host plant for this fungus. The current pathogenicity test showed that most *F. sylvatica* and all *A. saccharum* seedlings were able to limit the extension of wood discoloration, which prevented any further negative impact on the health of the seedlings. However, the high susceptibility of some beech seedlings turned out to be disturbing. It manifested as sudden wilting of the entire crown or the formation of extensive necrotic bark lesions. These results indicate that Polish isolates of *D. virescens* may pose a greater threat to *F. sylvatica* than to *A. saccharum*. There is a need for further research on the taxonomy and lifestyle of this fungus as well as an assessment of phytosanitary risk for the European forests. The rapid rate of production of ascospores and vegetative endoconidia by *D. virescens* as well as their huge abundance may be an important factor in the spread and in the establishment of the fungus in new areas. Moreover, based on the temperature assay, it can be concluded that the development of *D. virescens* mycelium can occur, although at different rates, throughout the year, excluding frosty winter periods.

## Figures and Tables

**Figure 1 jof-10-00465-f001:**
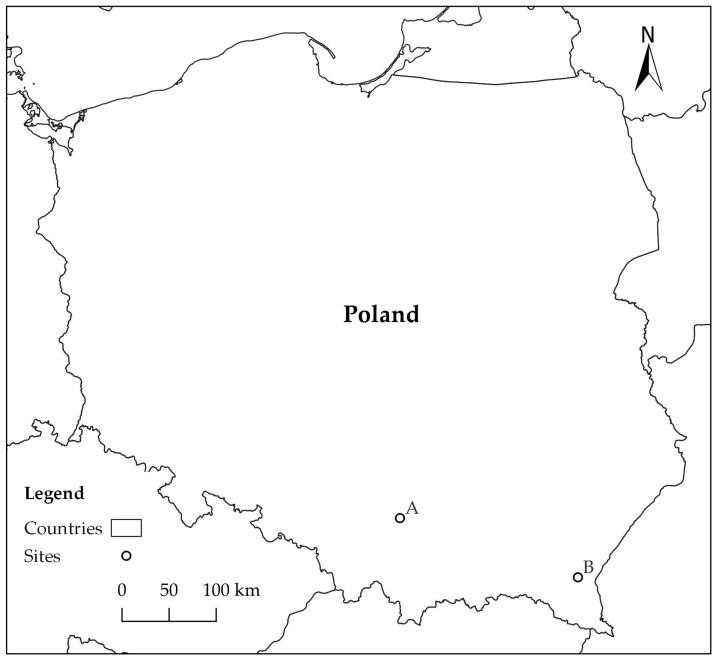
Location of forest sites: A—Ojców National Park, B—Rozpucie (Brzozów Forest District).

**Figure 2 jof-10-00465-f002:**
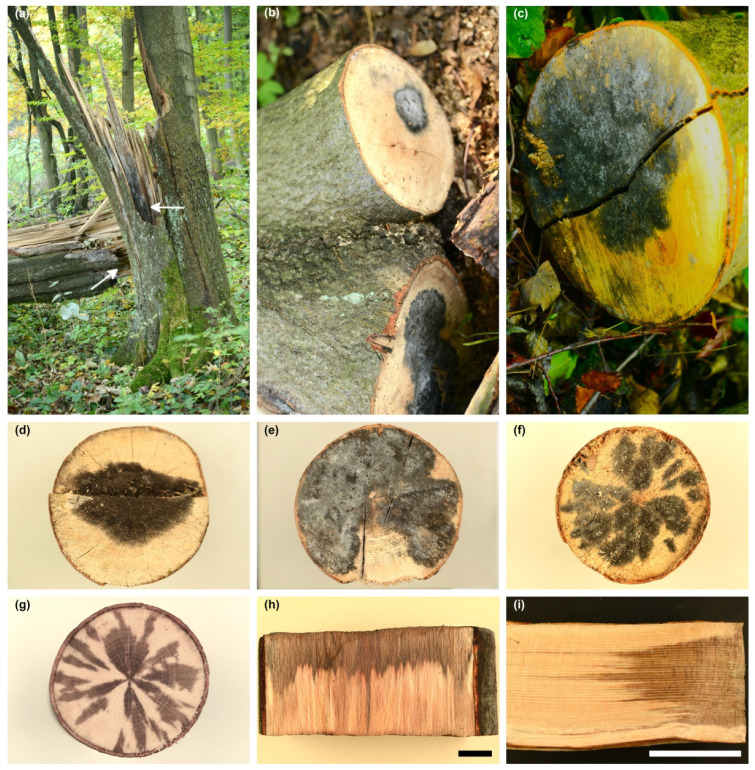
Occurrence of blackish fungal growth (BFG) on *Fagus sylvatica* on two sites in Poland; Site A: (**a**) *F. sylvatica* trunk broken by the wind; arrow shows BFG on the surface of wood without bark; (**b**,**c**) BFG on cross section of logs. Site B: (**d**,**e**) BFG, uniform structures on the face of the logs; (**f**) numerous small areas of BFG on the face of the log; (**g**) wood discoloration 3 cm below the symptoms shown in (**f**); (**h**,**i**) wood discoloration in logs in which face was covered with BFG; scale bars: (**h**) = 1 cm, (**i**) = 10 cm.

**Figure 3 jof-10-00465-f003:**
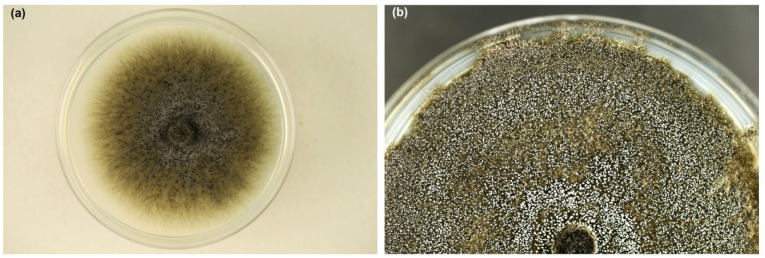
Colonies of *Davidsoniella virescens* on MEA at 20 °C: (**a**) one week old, (**b**) three weeks old (Petri dishes 9 cm in diameter).

**Figure 4 jof-10-00465-f004:**
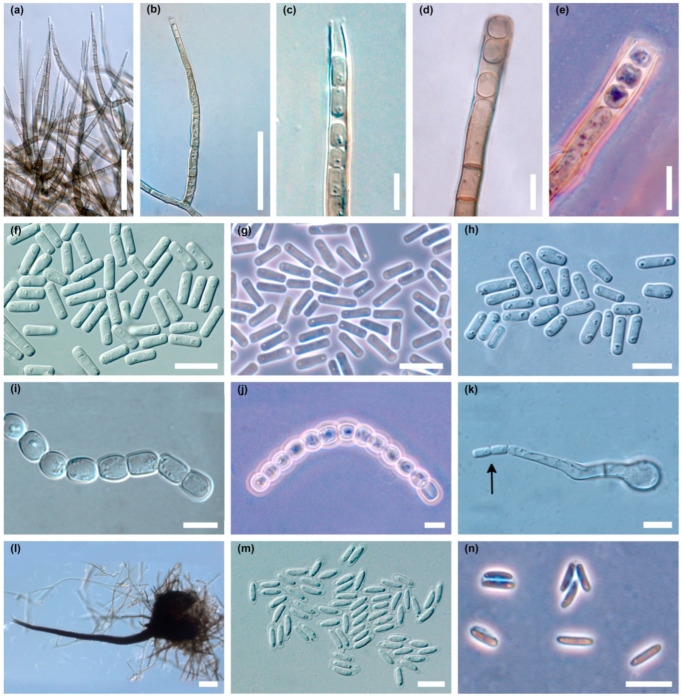
Microscopic structures of *Davidsoniella virescens* in vitro on MEA: (**a**) very numerous phialophores of type A produced in a young colony, (**b**) a single phialophore of type A, (**c**) apical part of the phialophore type A with endoconidia, (**d**) phialophore type B with endoconidia, (**e**) phialophore type B with endoconidia, phase contrast, (**f**,**g**) endoconidia type A in a young colony, (**f**) = DIC, (**g**) = phase contrast, (**h**) endoconidia type A in two-week-old colony, (**i**,**j**) chains of type B endoconidia, (**i**) = DIC, (**j**) = phase contrast, (**k**) germinating type B endoconidium producing endoconidia (arrow), (**l**) perithecium, long neck with divergent ostiolar hyphae, (**m**,**n**) ascospores surrounded by transparent sheath, (**m**) = DIC, (**n**) = phase contrast; scale bars: (**a**,**b**) = 50 µm, (**c**–**k**) = 10 µm, (**l**) = 50 µm, (**m**,**n**) = 10 µm.

**Figure 5 jof-10-00465-f005:**
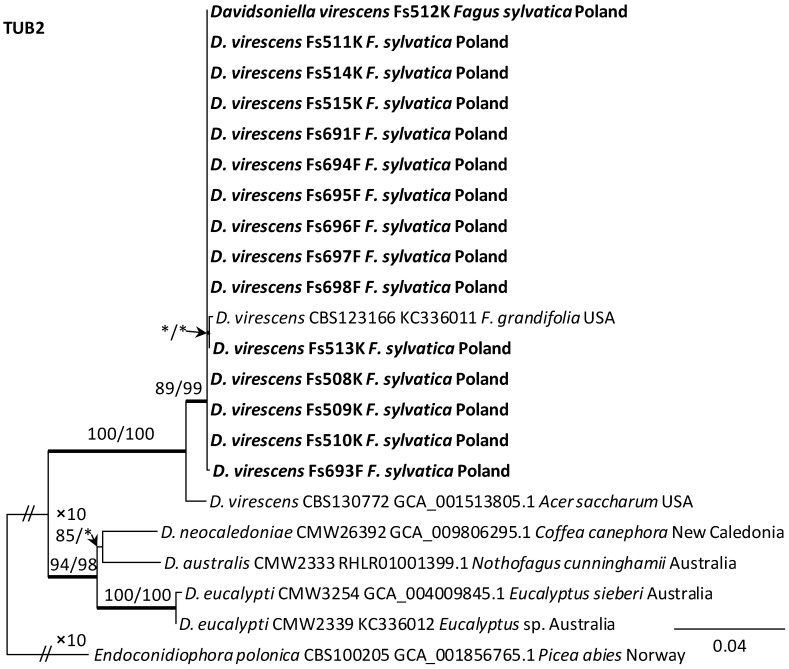
Phylogram obtained from analyses of the beta-tubulin (TUB2) data for the genus *Davidsoniella*. Sequences obtained during this study are presented in bold type. The presented phylogram was obtained from Maximum Likelihood (ML) analyses. The bootstrap values ≥ 75% for ML and Maximum Parsimony (MP) analyses are presented at nodes as follows: ML/MP. Bold branches indicate posterior probability values ≥ 0.95 obtained from Bayesian Inference (BI) analyses. * Bootstrap values < 75%. The tree is drawn to scale (see bar), with branch length measured in the number of substitutions per site. *Endoconidiophora polonica* represents the outgroup.

**Figure 6 jof-10-00465-f006:**
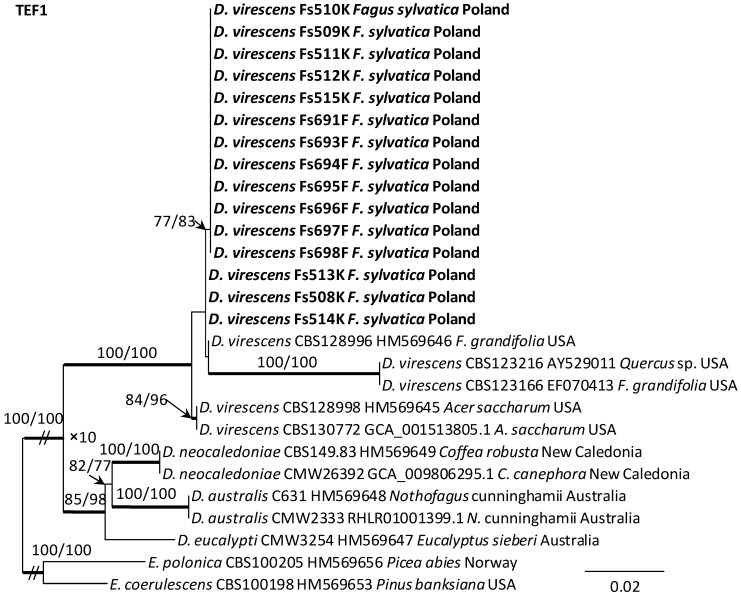
Phylogram obtained from analyses of the translation elongation factor 1-alpha (TEF1) data for the genus *Davidsoniella*. Sequences obtained during this study are presented in bold type. The presented phylogram was obtained from Maximum Likelihood (ML) analyses. The bootstrap values ≥ 75% for ML and Maximum Parsimony (MP) analyses are presented at nodes as follows: ML/MP. Bold branches indicate posterior probability values ≥ 0.95 obtained from Bayesian Inference (BI) analyses. The tree is drawn to scale (see bar), with branch length measured in the number of substitutions per site. *Endoconidiophora coerulescens* and *E. polonica* represent the outgroup.

**Figure 7 jof-10-00465-f007:**
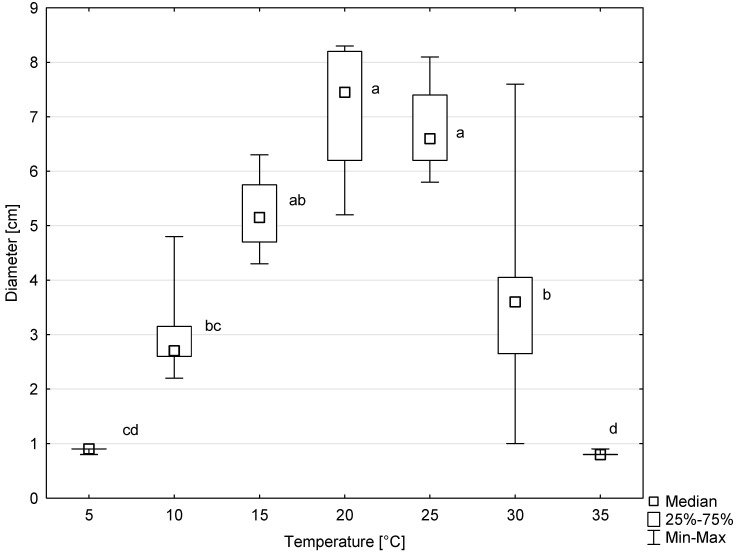
Colony diameters of six *Davidsoniella virescens* isolates after 8 days of growth at different temperatures. Values indicated with different letters were determined to be significantly different at α = 0.05 in Kruskal–Wallis test.

**Figure 8 jof-10-00465-f008:**
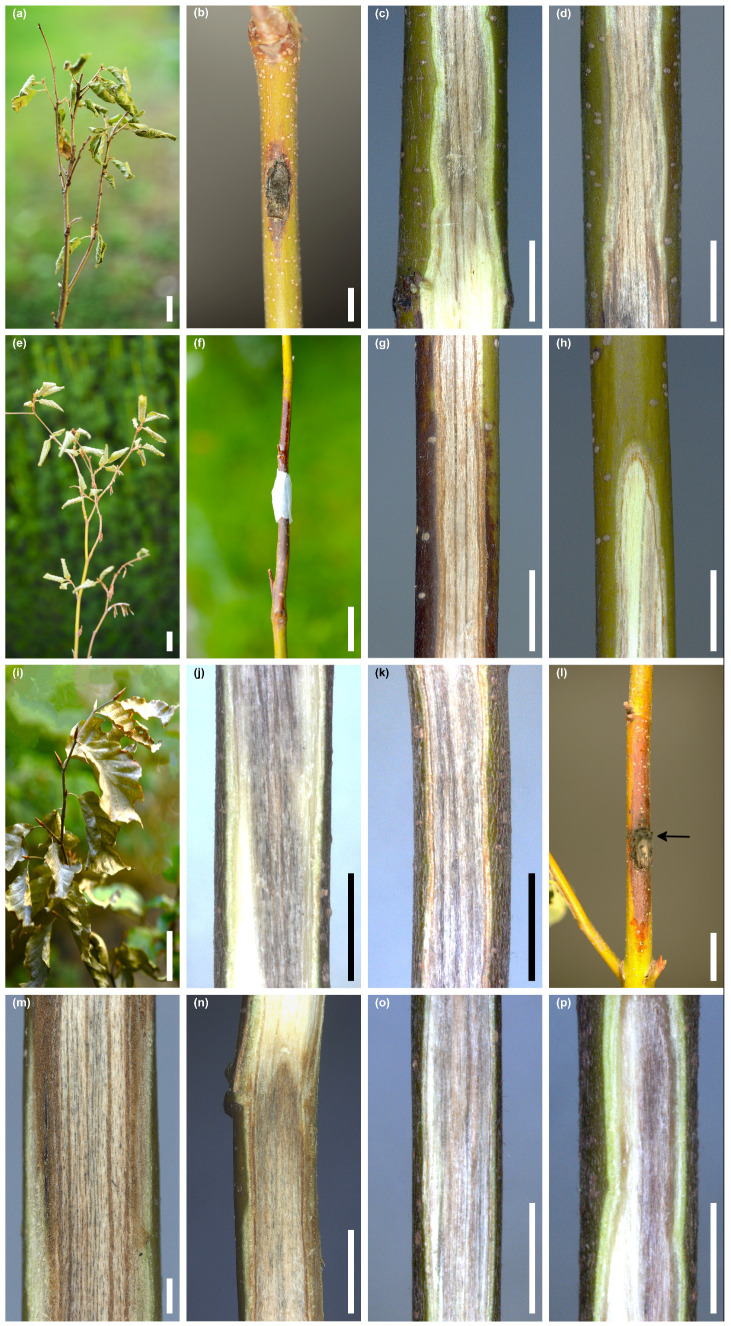
Symptoms in five *Fagus sylvatica* seedlings that manifested wilting 3–6 weeks post inoculation with *Davidsoniella virescens* (WD = wood discoloration): Seedling no. 5/Fs513F: (**a**) wilting symptoms, (**b**) inoculation site and local bark necrosis, (**c**) WD at the lower border with living tissue, inner bark is green, (**d**) WD up to 2.5 cm above the inoculation wound, inner bark is dying, Seedling no. 4/Fs697F: (**e**) wilting symptoms, (**f**) inoculation site and extended bark necrosis, (**g**) WD 3–5 cm above the inoculation wound, (**h**) streaky WD in the upper part of the stem, Seedling no. 2/Fs508K: (**i**) wilting symptoms, (**j**) WD 3–5 cm above the root collar, (**k**) WD 5–7 cm above the inoculation wound, Seedling no. 6/Fs697F: (**l**) inoculation site (arrow) and extended bark necrosis, (**m**) WD below the inoculation wound, (**n**) WD at the upper border with the living tissue, Seedlings no. 3/Fs509K: (**o**) WD 8–10 cm above the wound, (**p**) WD at the stem base, up to 2 cm above the root collar; scale bars: (**a**) = 50 mm, (**b**–**d**) = 5 mm, (**e**) = 50 mm, (**f**) = 20 mm, (**g**,**h**) = 5 mm, (**i**) = 50 mm, (**j**,**k**) = 5 mm, (**l**) = 10 mm, (**m**) = 1 mm, (**n**–**p**) = 5 mm.

**Figure 9 jof-10-00465-f009:**
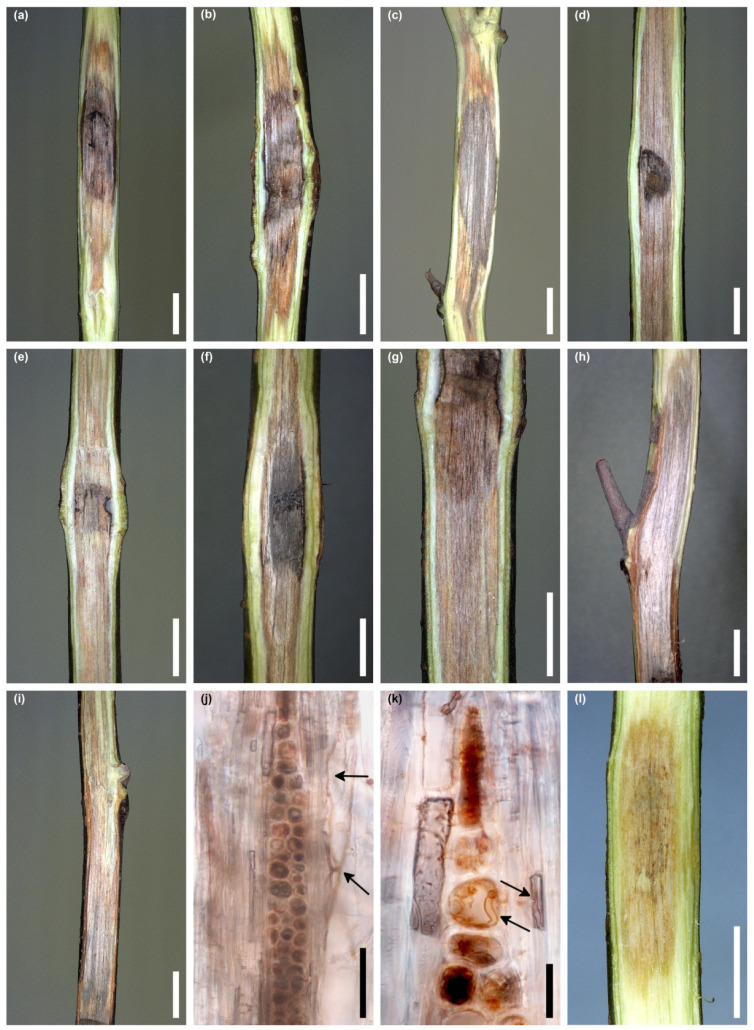
Internal symptoms in stems of *Fagus sylvatica* 4 months post inoculation with *Davidsoniella virescens* (WD = wood discoloration): (**a**–**c**) WD less than 5 cm long, (**d**–**g**) WD more than 5 cm long, (**h**,**i**) extensive grey-brown WD and dying of the bark above the inoculation site; in seedling no. 4/Fs508K (**h**) and no. 3/Fs513F (**i**), (**j**,**k**) hand-drawn tangential section through DW: *D. virescens* hyphae in the vessels, arrows (**j**), *D. virescens* hyphae in the vessels and in parenchyma cell, arrows (**k**), (**l**) WD in control seedling; scale bars: (**a**–**i**) = 5 mm, (**j**,**k**) = 50 µm, (**l**) = 5 mm.

**Figure 10 jof-10-00465-f010:**
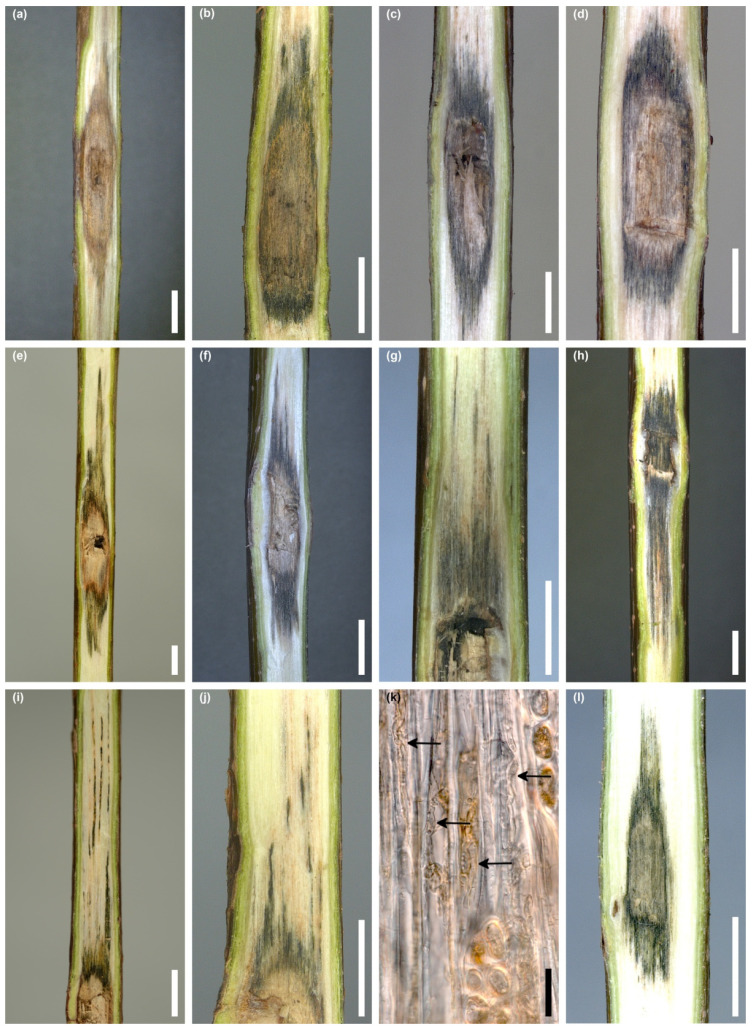
Internal symptoms in stems of *Acer saccharum* 4 months post inoculation with *Davidsoniella virescens* (WD = wood discoloration): (**a**) brown WD above and below the wound, (**b**–**d**) limited dark grey WD, (**e**–**h**) dark grey irregular WD with dark streaks, (**i**,**j**) WD in form of thin lines, (**k**) hand-drawn tangential section through discolored wood, *D. virescens* hyphae in the vessels (arrows), (**l**) WD in control seedling; scale bars: (**a**–**j**) = 5 mm, (**k**) = 50 µm, (**l**) = 5 mm.

**Figure 11 jof-10-00465-f011:**
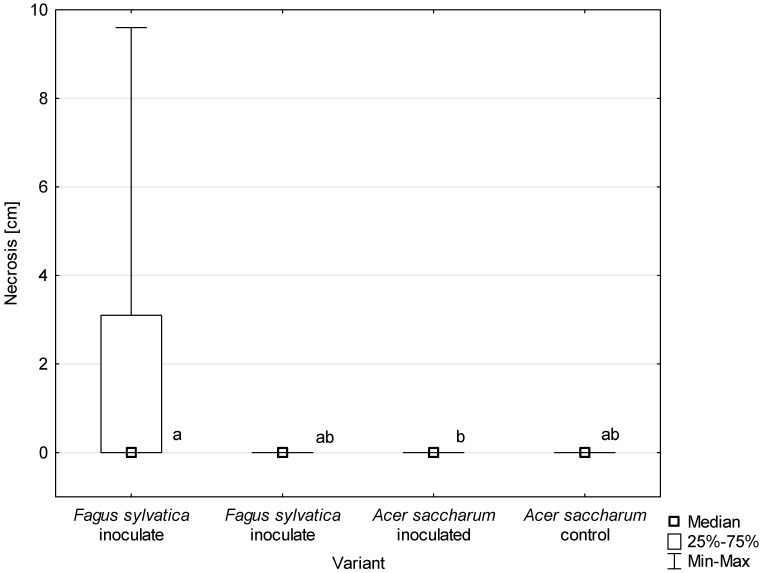
Length of bark necrosis on stems of tested trees 4 months post inoculation with *Davidsoniella virescens*. Data are presented in total for all tested isolates. Values indicated with different letters were determined to be significantly different at α = 0.05 in Kruskal–Wallis test.

**Table 1 jof-10-00465-t001:** *Davidsoniella virescens* isolates obtained from *Fagus sylvatica* wood and used in phylogenetic analyses.

Strain *	Origin ^1^	Collection Date	Collector ^2^	GenBank Accession Numbers		
				18S-ITS-28S	TUB2	TEF1
Fs508K *	Rozpucie	26 August 2021	PB	PP813732	PP826203	PP826220
Fs509K *	Rozpucie	26 August 2021	PB	PP813733	PP826204	PP826221
Fs510K	Rozpucie	26 August 2021	PB	PP813734	PP826205	PP826222
Fs511K *	Rozpucie	26 August 2021	PB	PP813735	PP826206	PP826223
Fs512K	Rozpucie	26 August 2021	PB	PP813736	PP826207	PP826224
Fs513K	Rozpucie	26 August 2021	PB	PP813737	PP826208	PP826225
Fs514K	Rozpucie	26 August 2021	PB	PP813738	PP826209	PP826226
Fs515K	Rozpucie	26 August 2021	PB	PP813739	PP826210	PP826227
Fs513F *	Ojców	21 August 2018	TK	PP813740	PP826211	PP826228
Fs514F	Ojców	21 August 2018	TK	PP813741	PP826212	PP826229
Fs691F	Ojców	10 October 2018	TK	PP813742	PP826213	PP826230
Fs693F	Ojców	10 October 2018	TK	PP813743	PP826214	PP826231
Fs694F	Ojców	10 October 2018	TK	PP813744	PP826215	PP826232
Fs695F	Ojców	10 October 2018	TK	PP813745	PP826216	PP826233
Fs696F	Ojców	10 October 2018	TK	PP813746	PP826217	PP826234
Fs697F *	Ojców	10 October 2018	TK	PP813747	PP826218	PP826235
Fs698F *	Ojców	10 October 2018	TK	PP813748	PP826219	PP826236

* Isolates used in temperature assay and in pathogenicity test; ^1^ Rozpucie—forest site B in the Brzozów Forest District, Ojców—forest site A in Ojców National Park; ^2^ PB—Piotr Bilański, TK—Tadeusz Kowalski.

**Table 2 jof-10-00465-t002:** Symptoms in five *Fagus sylvatica* seedlings that manifested wilting 3–6 weeks post inoculation with *Davidsoniella virescens*.

No. of Seedling/ Isolate	Wilting Post Inoculation	Symptoms on Leaves	Length of Bark Necrosis (cm)	Axial Wood Discoloration (cm)	Wood Discoloration, Stem/Roots
5/Fs513F	3 weeks	most of the leaves pale green, occasionally brown, and curled from the edges (Figure 8a)	1.6 (Figure 8b)	7.6; grey-brown with darker longitudinal lines (Figure 8c,d)	stem
4/Fs697F	3 weeks	all leaves greyish-green, slightly wrinkled, curling from the edges (Figure 8e)	9.6 (Figure 8f)	16.1; grey-brown with darker longitudinal lines (Figure 8g) or streaks (in upper part) (Figure 8h)	stem
2/Fs508K	4 weeks	leaves grey-green, brown at the edges, bent, starting to curl at the edges (Figure 8i)	4.2 (perith. *)	17.0; grey-brown with darker longitudinal lines (Figure 8j,k)	stem
6/Fs697F	5 weeks	leaves greyish-green, wrinkled and curled from the edges	5.8 (Figure 8l)	7.1; grey-brown with darker longitudinal lines, dark grey in places (Figure 8m,n)	stem
3/Fs509K	6 weeks	leaves grey-green, wrinkled, some of them curled at the edges	8.0 (perith. *)	26.0; grey-brown with darker longitudinal lines (Figure 8o); at the stem base, wood discolored in a striped form (Figure 8p) extending to the roots	stem, roots

*—numerous mature perithecia in place of inoculation.

**Table 3 jof-10-00465-t003:** Presence of bark necrosis (necr.) and discoloration (discol.) in stem wood of *Fagus sylvatica* and *Acer saccharum* 4 ^1^ months post inoculation with *Davidsoniella virescens*.

Isolate Number/Symptom		Length ^2^ of Wood Discoloration (cm)						Mean (cm) ^3,4^	Standard Deviation	Standard Error of Mean
		Stem Number								
		1	2	3	4	5	6			
*Fagus sylvatica*										
Fs698F	discol.	14.5	9.6	11.8	2.6	4.6	7.7	8.47 ^a^	4.44	1.81
	necr.	-	-	-	-	-	-	0.00 ^A^	0.00	0.00
Fs697F	discol.	2.6	4.1	3.2	16.1 *	3.1	7.1 *	6.03 ^ab^	5.19	2.12
	necr.	-	1.8	-	9.6	-	5.8	2.87 ^A^	3.99	1.63
Fs513F	discol.	6.4	9.1	11.4	7.8	7.6 *	5.2	7.92 ^a^	2.16	0.88
	necr.	-	-	7.2	-	1.6	-	1.47 ^A^	2.88	1.18
Fs508K	discol.	7.8	17.0 *	7.4	8.4	10.1	14.6	10.88 ^a^	3.99	1.63
	necr.	-	4.2	3.6	3.4	2.8	-	2.33 ^a^	1.86	0.76
Fs509K	discol.	8.0	9.2	26.0 *	7.8	8.5	5.5	10.83 ^a^	7.53	3.08
	necr.	-	-	8.0	1.3	1.6	-	1.82 ^A^	3.11	1.27
Fs511K	discol.	15.6	10.4	8.3	5.1	7.0	8.4	9.13 ^a^	3.62	1.48
	necr.	3.4	3.5	-	2.3	-	1.7	1.82 ^A^	1.56	0.64
control	discol.	1.4	1.1	1.3	2.2	1.7	1.3	1.50 ^c^	0.39	0.16
	necr.	-	-	-	-	-	-	0.00 ^A^	0.00	0.00
*Acer saccharum*										
Fs698F	discol.	1.8	3.2	1.4	1.8	2.8	1.6	2.10 ^bc^	0.72	0.30
Fs697F	discol.	1.6	1.3	3.0	2.2	1.4	3.3	2.13 ^bc^	0.85	0.35
Fs513F	discol.	1.8	1.2	3.0	1.4	3.4	1.7	2.08 ^bc^	0.90	0.37
Fs508K	discol.	1.3	1.5	1.6	6.1	1.2	1.3	2.17 ^c^	1.93	0.79
Fs509K	discol.	2.3	1.9	5.0	1.6	1.8	4.8	2.90 ^bc^	1.57	0.64
Fs511K	discol.	1.4	2.0	2.5	3.4	2.1	1.3	2.12 ^bc^	0.77	0.32
control	discol.	1.6	0.9	1.1	1.0	1.2	1.8	1.27 ^c^	0.36	0.15

^1^ The period for plants with wilting symptoms (*) was 3–6 weeks. ^2^ Length includes dense, striped or linear discoloration of the wood. ^3^ Values designated by different small letters differed significantly (*p* < 0.05) according to the analysis of variance (ANOVA) and post hoc Tukey HSD test. ^4^ Values marked with the same capital letters were not significantly different (*p* > 0.05) according to multiple comparisons of mean ranks for all groups of variables following the Kruskal–Wallis test.

**Table 4 jof-10-00465-t004:** Results of *Davidsoniella virescens* re-isolation 4 months after inoculation of *Fagus sylvatica* and *Acer saccharum* seedlings (from *F. sylvatica* seedlings with wilting symptoms, re-isolations were performed 3–6 weeks after inoculation).

Site of Sampling of Wood Fragments	*Fagus sylvatica*				*Acer saccharum*	
	With Wilting Symptoms		Without Wilting Symptoms		Number of Wood Fragments Used for Isolation	Number (%) of Wood Fragments Which Yielded *D. virescens*
	Number of Wood Fragments Used for Isolation	Number (%) of Wood Fragments Which Yielded *D. virescens*	Number of Wood Fragments Used for Isolation	Number (%) of Wood Fragments Which Yielded *D. virescens*		
Inoculation wound	36	0 (0.0)	156	15 (9.6)	150	1 (0.7)
Zone slightly above and below the wound	36	11 (30.6)	190	18 (9.5)	-	-
Lower border of discoloration	36	26 (72.2)	179	41 (22.9)	215	2 (0.9)
Upper border of discoloration	30	19 (63.3)	190	62 (32.6)	219	3 (1.4)
Total	138	56 (40.6)	715	136 (19.0)	584	6 (1.0)
Number (%) of seedlings	5	5 (100.0)	31	26 (83.9)	36	5 (13.9)
Number of control seedlings (wood fragments)	-	-	6 (36)	0 (0.0)	6 (36)	0 (0.0)

## Data Availability

The data presented in this study are available in Appendix A or deposited in publicly available databases such as GenBank and TreeBASE.

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
