# Peer review of "Recognition of Davidsoniella virescens on Fagus sylvatica Wood in Poland and Assessment of Its Pathogenicity"

_jof, 2024, doi:10.3390/jof10070465_

Round 1

Reviewer 1 Report

1. more similar symptoms between natural infections ( fig.2) and inoculations (fig. 8) are suggeated to add. 

2. More information should be provided for inoculations, spores or mylelial plugs?

3. temperature assess only for mylelial growth, why not sporulation,germination and pathogenicity?

4. I suggested all isolates be confirm of pathogencity firstly , then for morphorlogy and phylogenetic and  ----

1. ITS is not gene. Pay attention to the writing of gene.

2. the conclusions is too long. 

3. same scale bars used for similar circumstances.

Author Response

Thank you for your positive assessment of our research work. We have taken your suggestions into account in the manuscript whenever possible; we can only refer to some of them through explanation.

Point 1: More similar symptoms between natural infections (fig. 2) and inoculations (fig. 8) are suggested to add.

Response 1: We added the text in line 525/first version

Mycelium and sporulation structures of D. virescens did not form on the bark of the currently analyzed logs. The same was observed in inoculated plant seedlings, in which only occasionally (11.1% of beech, 5.6% of sugar maple) perithecia developed at the place of injury. Regardless of whether colonization occurred naturally (logs) or as a result of artificial inoculation (seedlings), the wood showed symptoms of gray or grey-brown discoloration, which was uniform or marked with dark streaks.

Point 2: More information should be provided for inoculations, spores or mycelial plugs?

Response 2: We added the text in line 225/first version

These beech-wood pieces used for inoculation were overgrown by mycelium along with endoconidia and ascospores and the structures producing them (phialophores, perithecia).

Point 3: Temperature assess only for mycelial growth, why not sporulation, germination and pathogenicity?

Response 3: In previous analyzes on the threat of D. virescens to European forests, much attention was paid to its occurrence in North America depending on temperature (Webber 2008, Jeger et al. 2017, see text, lines 614-618/first version). This was the reason why this aspect was included as one of the goals in the current work. If this fungus is discovered in other countries and begins to play a significant phytosanitary role in Europe, phytopathologists in various countries will certainly undertake extensive research to clarify various aspects of the biology and lifestyle of this fungus.

Point 4: I suggested all isolates be confirm of pathogencity firstly, then for morphorlogy and phylogenetic.

Response 4: We conducted research according to current standards common in mycological and phytopathological works published in recognized scientific journals. We performed morphological analysis of fungal structures in situ and in vitro as well as multi-locus molecular characterization and phylogenetic analysis, which allowed us to identify the fungal species. After it turned out to be D. virescens, previously unknown in Europe, the results could be published already at this stage. However, we additionally assessed pathogenicity. It is common practice that the pathogenicity test is performed on a representative number of samples, because it is an extremely labor-intensive experiment (plant material, inoculation, evaluation, reisolation). Only because we had previously identified the fungus species, we included Acer saccharum in the pathogenicity tests.

Point 5:  ITS is not gene. Pay attention to the writing of gene.

Response 5: In the narrow sense of the term, we agree with the statement that ITS is not a gene. However, in the broad sense of the term, ITS is considered by some authors to be a gene. The title of the publication may serve as an example of the latter view of the term: “Internal Transcribed Spacer (ITS) gene as an accurate DNA barcode for identification of macroscopic fungus in Aceh https://doi.org/10.13057/biodiv/d230514”. We have therefore decided not to change the terminology used in the work regarding ITS.

Point 6:The conclusions is too long.

Response 6: This section has been shortened by approx. 40%.

Point 7: Same scale bars used for similar circumstances.

Response 7: In our opinion, the same scale bars cannot be used in similar circumstances because the bars will either be too short or too long. The current layout of the components of our figures is a compromise between the same scale and image legibility. We kindly ask you to accept the scale bars so presented.

Reviewer 2 Report

The manuscript titled “Recognition of Davidsoniella virescens on Fagus sylvatica Wood in Poland and Assessment of its Pathogenicity” is devoted to the first finding of  Davidsoniella virescens outsides of North America.

In South Poland several cultures of this fungus were obtained. All 10 cultures produced an intense sweet odor. This fungus both in situ and in vitro produced abundantly perithecia with long necks and asexual stage. The nucleotide sequences for five gene fragments of representative cultures were used in phylogenetic analyses: 18S; the internal transcribed spacer regions ITS1 and ITS2, including the 5.8S gene 15 (ITS); 28S region of the ribosomal RNA (rRNA), β-tubulin 2 (TUB2) and translation elongation 16 factor 1-α (TEF1). Based on morphological and phylogenetic analyses, the fungus on European beech in Poland was identified as Davidsoniella virescens.

Six D. virescens isolates were used for pathogenicity assay. They were inoculated into wounds on stems of two-year-old seedlings of Fagus sylvatica and Acer saccharum, and final evaluation was performed four months after inoculation. 13.9% of F. sylvatica seedlings showed wilting symptoms throughout the entire crown within 3–6 weeks after inoculation. Moreover, after 4  months on the stems of nearly 1/3 of inoculated beech seedlings, necrotic lesion was formed, without any symptoms of wilting. Acer saccharum  plants were less affected. Pathogenicity test showed that the D. virescens isolates identified in South Poland may pose a threat to native European beech.

The manuscript is well-written and respresents accurate investigation of new for Europe pathogen of highly important trees. 

The manuscript is accurate and well-written. No other comments

Author Response

Thank you for your positive assessment of our research work.

Reviewer 3 Report

The paper entitled 'Recogintion of Davidsoniella virescens on Fagus sylvatica wood in Poland and assessment of its pathogenicity' describes the first observation of Davidsoniella virescens fungus outside the North America. The result is supported by a comprehensive study, including moprphological, phylogenetic and pathogenicity assays. The methods are described well, and all the procedures can be reproduced. 

The single major comment is: I would like the authors to give more information about phylogenetic features of the sequences were used (variable, phylogenetically informative positions etc). What marker was the most phylogenetically robust? Why are the trees based on b-TUB and TEF sequences represented in the MS tex and the tree based on the concatenated rDNA sequence is not?  

Line 24: F. sylvatica should be in Italic

Line 36: species name of the fungus under study should be among the Keywords

Line 45: probably, '....this fungus possesses growth and morhological features which are very similar.....' would be better.

lines 335-336: d

roplets

Lines 376, 386 The genes (bTUB, TEF1a) names under the figures are indicated in Italic, but in the tex are not

Author Response

Thank you for your positive assessment of our research work.

Point 1: I would like the authors to give more information about phylogenetic features of the sequences were used (variable, phylogenetically informative positions etc).

Response 1: . We added a single sentence “The datasets used in the phylogenetic analysis contained different number of variable characters: 18S - 5, ITS - 59, 28S - 20; TUB2 - 298 and TEF1 - 273, of which 3, 51, 13, 108 and 219 were parsimoniously informative, respectively”.

Point 2: What marker was the most phylogenetically robust?

Response 2: We added a single sentence “TEF1 was the most phylogenetically robust among all markers tested (Figure 6)”.

Point 3: Why are the trees based on b-TUB and TEF sequences represented in the MS text and the tree based on the concatenated rDNA sequence is not?

Response 3: We analysed five DNA fragments and had to transfer some of the phylogenetic trees to supplementary material to avoid a large number of figures in the text of the manuscript. The tree based on the 28S data had a similar topology to the TUB2 and TEF1 trees but the latter had more strongly statistically supported clades. Therefore, we chose to present only the results of protein-coding gene analyses in the body of the text.

Point 4: Line 24: F. sylvatica should be in Italic.

Response 4: F. sylvatica has been corrected to italics.

Point 5: Line 36: species name of the fungus under study should be among the Keywords.

Response 5: we have done it

Point 6: Line 45: probably, '....this fungus possesses growth and morhological features which are very similar.....' would be better.

Response 6: we changed the text as suggested

Point 7: Lines 335-336: droplets.

Response 7: We have corrected the erroneously split word "droplets".

Point 8: Lines 376, 386 The genes (bTUB, TEF1a) names under the figures are indicated in Italic, but in the text are not.

Response 8: We have removed italics from gene names in figure captions.

Reviewer 4 Report

The authors in their manuscript entitled “Recognition of Davidsoniella virescens on Fagus sylvatica Wood in Poland and Assessment of its Pathogenicity” present a well-documented first report of the fungal species Davidsoniella virescens. The methodology and technical approaches are standard, Koch’s postulates are covered, and the multi-locus molecular characterization followed is informative regarding phylogenetic relations (other territories) and conclusive regarding the species. There are no major comments on the work nor for the presentation. Only some trivial comments regarding a putative modification of the title for paragraph 2.3 (refers to growth thus probably a title referring to temperature-related colony growth than temperature assays could be decided by the authors, line 219 (change the word “decade” to “first ten-days” (I suppose...), lines 335-336 (typographic).

However, in the context of an informal discussion I would like to ask the authors if they have a speculation about how the pathogen entered the country and if they have related information from neighboring countries. Is there a possibility that the species existed/exists in a quiescent state as an endophyte, somehow activated due to a reason? It is a wild forest I presume. Were there any tree plantations before? You report that the age of the trees is around 90 years. Any idea how it entered the territory?

The first isolation goes back to 2018. I suppose that the Polish state, NRL, and EURL are aware of the pathogen presence since it is considered a quarantine pest (A1 Quarantine pest (Annex II A) 2019). Any further guidance regarding official surveys? Do you see any connection with C.platani, and B. fagacearum presence?

Thank you.

Please refer to main comments. There are no corrections-ammendments needed.

Author Response

Thank you for your positive assessment of our research work.

Point 1: A putative modification of the title for paragraph 2.3 (refers to growth thus probably a title referring to temperature-related colony growth than temperature assays could be decided by the authors.

Response 1: we changed the text as suggested

Point 2: Line 219 (change the word “decade” to “first ten-days” (I suppose...).

Response 2: we have changed the text

Point 3: Lines 335-336 (typographic)

Response 3: We have corrected the erroneously split word "droplets".

Point 4: However, in the context of an informal discussion I would like to ask the authors if they have a speculation about how the pathogen entered the country and if they have related information from neighboring countries. Is there a possibility that the species existed/exists in a quiescent state as an endophyte, somehow activated due to a reason? It is a wild forest I presume. Were there any tree plantations before? You report that the age of the trees is around 90 years. Any idea how it entered the territory?

Response 4: The indicated aspects are very important. However, to answer them, we need to have much more information. It is most likely that the fungus was accidentally brought along with wood or other plant substrate from North America. We point to this possibility in the manuscript (Lines 82-83/first version). Both in Poland and in other European countries, a survey of tree stands for the presence of D. virescens is necessary. This may help determine the route of this fungal species to Europe. Both sites in Poland where D. virescens was found have had forests for a long time (site A is within the National Park).Fungi from the Ceratocystidaceae family are rarely reported as endophytes.Recently, there has been a lot of research on endophytes, if it was present, it would have been discovered, especially since the sequences of this species are deposited in GenBank.

Point 5: The first isolation goes back to 2018. I suppose that the Polish state, NRL, and EURL are aware of the pathogen presence since it is considered a quarantine pest (A1 Quarantine pest (Annex II A) 2019). Any further guidance regarding official surveys? Do you see any connection with C. platani, and B. fagacearum presence?

Response 5: We see no connection with C. platani, and B. fagacearum presence. However, the mode of action of D. virescens inside the trunks of infected trees may be similar to that of abovementioned fungi (Line 637- 646/first version). We point to the problem of D. virescens as quarantine pest in manuscript (Line 89-97/first version). This problem is unclear, so far it is not certain whether the saprotrophic forms occurring on various hardwoods in the USA are the same species as the causal factor of sapstreak diseases on sugar maple trees [24]. The EU has not clearly defined whether all strains of D. virescens are under quarantine. Indirectly, from the various restrictions it can be concluded that these are strains of Acer saccharum. In the current situation, the relevant phytosanitary services will have to take a clear position and possibly, through their structures operating in all countries, take some initiatives to analyze the occurrence of D. virescens. We, as academics, can provide substantive advice and assistance if asked. It is difficult to predict how the situation will develop. As we know, some 'alien fungi' turned out to be very dangerous (e.g. Hymenoscyphus fraxineus, Ophiostoma novo-ulmi), others not yet, despite their spread in Europe (e.g. Eutypella parasitica).

Reviewer 5 Report

The paper presents the identification of Davidsoniella virescens in Poland on the wood of common beech. The report is of the great interest because it is the novel and first report of this pathogen outside North America. This study is comprehensive and a great addition to the previous reports of this pathogen and has brought new knowledge on D. virescens. It may encourage surveys and possible detection of this pathogen in other European countries.

The study includes different methods for detection and identification (isolation, morphology, microscopy, temperature assay, molecular tests, sequencing, phylogeny) and pathogenicity tests on two different species (common beech and sugar maple).

The material and methodology section are presented in detail. I do not have any remarks for this section.

Results are presented properly followed by discussion section.

Conclusions are consistent with presented results.

References are relevant for the research in this area.

Figures and tables are illustrative and well presented. I emphasize the quality of the photos that vividly describe everything that is written in the text.

I do not have any negative remarks regarding the manuscript.

Author Response

(The authors gave the same response as above.)

Round 2

Reviewer 1 Report

I suggest to accept this revised ms.

I suggest to accept this revised ms.